# Quantitative Phase Analysis by X-ray Diffraction—Doping Methods and Applications

**Stanko Popović [1,2]**

[1]  Croatian Academy of Sciences and Arts, 10000 Zagreb, Croatia; spopovic@phy.hr
[2]  Department of Physics, Faculty of Science, University of Zagreb, 10000 Zagreb, Croatia

**Abstract:** X-ray powder diffraction is an ideal technique for the quantitative analysis of a multiphase sample. The intensities of diffraction lines of a phase in a multiphase sample are proportional to the phase fraction and the quantitative analysis can be obtained if the correction for the absorption of X-rays in the sample is performed. Simple procedures of quantitative X-ray diffraction phase analysis of a multiphase sample are presented. The matrix-flushing method, with the application of reference intensities, yields the relationship between the intensity and phase fraction free from the absorption effect, thus, shunting calibration curves or internal standard procedures. Special attention is paid to the doping methods: (i) simultaneous determination of the fractions of several phases using a single doping and (ii) determination of the fraction of the dominant phase. The conditions to minimize systematic errors are discussed. The problem of overlapping of diffraction lines can be overcome by combining the doping method (i) and the individual profile fitting method, thus performing the quantitative phase analysis without the reference to structural models of particular phases. Recent suggestions in quantitative phase analysis are quoted, e.g., in study of the decomposition of supersaturated solid solutions—intermetallic alloys. Round Robin on Quantitative Phase Analysis, organized by the IUCr Commission on Powder Diffraction, is discussed shortly. The doping methods have been applied in various studies, e.g., phase transitions in titanium dioxide, biomineralization processes, and phases in intermetallic oxide systems and intermetallic alloys.

**Keywords:** X-ray powder diffraction; quantitative phase analysis; matrix-flushing method; doping method; Rietveld method; intermetallic alloys; intermetallic oxides; biominerals

## 1. Introduction

All of the information relating to the crystal structure and microstructure and, particularly, to the quantitative composition of a multiphase sample, is stored in its X-ray diffraction pattern. It has been the goal of X-ray diffraction scientists since the discovery of X-rays to decode this information directly from the X-ray diffraction pattern. The elemental composition of a multiphase sample can be determined by chemical and spectroscopic techniques. However, by these techniques the chemical identity of crystalline phases and the fractions of the phases cannot be obtained in most cases. X-ray powder diffraction has been proved to be an ideal technique for the quantitative phase analysis (QPA) of a multiphase sample. The intensities of diffraction lines of a given phase are proportional to its (mass, molar) fraction and the QPA can be performed after the application of the correction for the absorption of X-rays in the sample [1,2]. A comprehensive description of QPA is given by Madsen, Scarlett, Kleeberg, and Knorr in Chapter 3.9 of the International Tables for Crystallography, Volume H: Powder Diffraction [3].

## 2. Essential Points in Short

Let a sample consist of several phases denoted with capital letters and let the same notation represent their (mass, molar) fractions:

$$A + B + C + \ldots + X + Y + \ldots = 1. \tag{1}$$

The integrated intensity of a selected diffraction line of a phase, say $A$, is related to its fraction:

$$I_A = K_A \, A/(d_A \, \mu) \tag{2}$$

$d_A$ being the density of the phase $A$, $\mu$ the mass absorption coefficient of the sample, and $K_A$ a factor depending on the nature of the specimen dependent effects (preferred orientation, grain size and shape, and extinction) and of the phase $A$, on the selected diffraction line and the geometry of the diffractometer.

For pure phase $A$, since $A = 1$, (2) changes into:

$$I_{A0} = K_A/(d_A \, \mu_A) \tag{3}$$

where $\mu_A$ is the mass absorption coefficient of the phase $A$. From (2) and (3) it follows:

$$I_A/I_{A0} = A \, (\mu_A/\mu). \tag{4}$$

Similarly, for another phase, say $B$, it follows:

$$I_B/I_{B0} = B(\mu_B/\mu). \tag{5}$$

From (4) and (5) one obtains:

$$A/B = K_{AB} \, (I_A/I_B), \; K_{AB} = (\mu_B/\mu_A)(I_{B0}/I_{A0}), \tag{6}$$

where $K_{AB}$ is a constant for the two phases which are considered, for the selected diffraction lines and for the diffractometer which is used. A direct application of (6) is not straightforward, since the absorption coefficients are not accurately known.

In order to solve the absorption problem, a semi-empirical internal standard method can be used. For each phase, say $A$, the fraction of which is to be determined, a calibration curve is constructed in such a way as to relate $A/S$ vs. $I_A/I_S$, where $S$ denotes the internal standard. According to (6), the calibration curve is a straight line having the slope $K_{AS}$. The slope is obtained from the intensity measurement of a series of mixtures with known ratios $A/S$. In order to find the fraction of the phase $A$ in the sample, a known fraction of the standard $S$ is added to the sample, the intensity ratio $I_A/I_S$ is measured and $A$ is found from the previously constructed calibration curve. A detailed description of the internal standard method, as applied in special cases, can be found in the textbook of Klug and Alexander [2] and in Reference [3] (pp. 346–347, 350–351).

The problem of the absorption of X-rays in the sample can be overcome by the matrix flushing method [4,5] and the doping methods [6–10].

## 3. Chung's Matrix Flushing Method

In the method developed by Chung [4,5], no calibration curve is needed, as the matrix effects—absorption coefficients—are flushed out of the intensity–fraction equation. This method is simpler and faster than the conventional internal-standard procedure. The method is based on the previous knowledge, or measurement, of relative-reference intensities of (the strongest) diffraction lines for each pair of phases that are present in the sample, or rather for each phase present in the sample and a reference phase (corundum, $\alpha$-$Al_2O_3$) in a binary mixture of a one-to-one mass/molar ratio.

The intensity–fraction relationship between each pair of phases in a multi-phase sample is not perturbed by the presence or absence of other phases. Contrary to most theoretical approaches, the working equations are simple and no complicated calculations are involved. All information related to the quantitative composition of the sample can be decoded directly from its diffraction pattern [4,5]. A concise description of Chung's method can be found in Reference [3], Sections: Reference Intensity Ratio Methods, Matrix-Flushing Method (pp. 347–348, 351–352).

## 4. Doping Methods

Doping methods are not described in Reference [3]. A detailed description of the doping methods is given in references [6–10]. Essential points from the references [6–10] are given here.

The doping methods involve the addition, to the original sample, of known amount(s) of the phase(s), the fraction of which is (are) to be determined. The corresponding equations, deduced with no approximation, relate the fraction of the phase to be determined to the intensities diffracted by that phase and by any non-added phase (reference phase) which is present in the sample, before and after doping. The intensity–fraction equations are free of the matrix effects—absorption coefficients. The methods can be applied to a sample containing unidentified phase(s), in simultaneous analysis only for the phases of interest, and in determination of the fraction of the amorphous content in the case when the fractions of all crystalline phases have been determined. Two doping methods are described here:

(i) simultaneous determination of the fractions of several phases using a single doping (This method has been considered by PANALYTICAL: In the description of the software of PANALYTICAL - *X'PERT QUANTIFY* it is stated: Addition (i.e., doping) models in which after the initial measurement on a sample, the sample is measured again with the concentration of the component of interest enriched (i.e., doped) by a known amount.);

(ii) determination of the fraction of the dominant phase**;**

### 4.1. Simultaneous Determination of the Fractions of Several Phases Using A Single Doping

The basic points can be summarized as follows. Let the sample consist of **N** phases. Its diffraction pattern is taken and a partial or complete identification of the phases is performed. The prominent, non-overlapping (in principle, the strongest) diffraction lines of particular phases are chosen and their net integrated intensities are measured. Then the sample is doped by known fractions of $M$ phases, the original fractions of which are to be determined ($M = 1, 2, 3, ... N - 1$). The intensities of the chosen diffraction lines are measured again. In principle, if $N - 1$ phases are added, the original fractions of all $N$ phases can be found.

The composition of the original sample is given by (1). If one wants to determine the fraction of the phase $X$, a known fraction of that phase, $X_a$, is added to the original sample. Then any other phase, say $Y$, may be used as the reference phase. For the doped sample the following is valid:

$$A_d + B_d + C_d + \ldots + X_d + X_a + Y_d + \ldots = 1. \tag{7}$$

It has been shown [6–8] that the fraction of the phase $X$ in the original sample is given by equation:

$$X = X_a R_{YX}/(P - P R_{YX}). \tag{8}$$

$P$ equals the fraction of the original composition in the doped sample, or, 1 – the total fraction of all the added phases in the doped sample. $R_{YX}$ is expressed through the intensities of the phases $X$ and $Y$ before ($I_X, I_Y$) and after ($I_{X\,d+a}, I_{Y\,d}$) doping:

$$R_{YX} = (I_{Y\,d}/I_{X\,d+a})(I_X/I_Y). \tag{9}$$

Several pairs of diffraction lines of the phases $X$ and $Y$ can be utilized in order to increase the accuracy of the measurement. Namely, from (8) and (9) it follows:

$$I_{X\,d+a}/I_{Y\,d} = K(I_X/I_Y). \tag{10}$$

The plot of $I_{X\,d+a}/I_{Y\,d}$ as a function of $(I_X/I_Y)$ is a straight line with the slope:

$$K = (X\,P + X_a)\,/(X\,P).$$

It follows that the fraction of the phase of interest, $X$, is given by:

$$X = X_a/(PK - P).$$

An example of the doping method is shown in Figure 1.

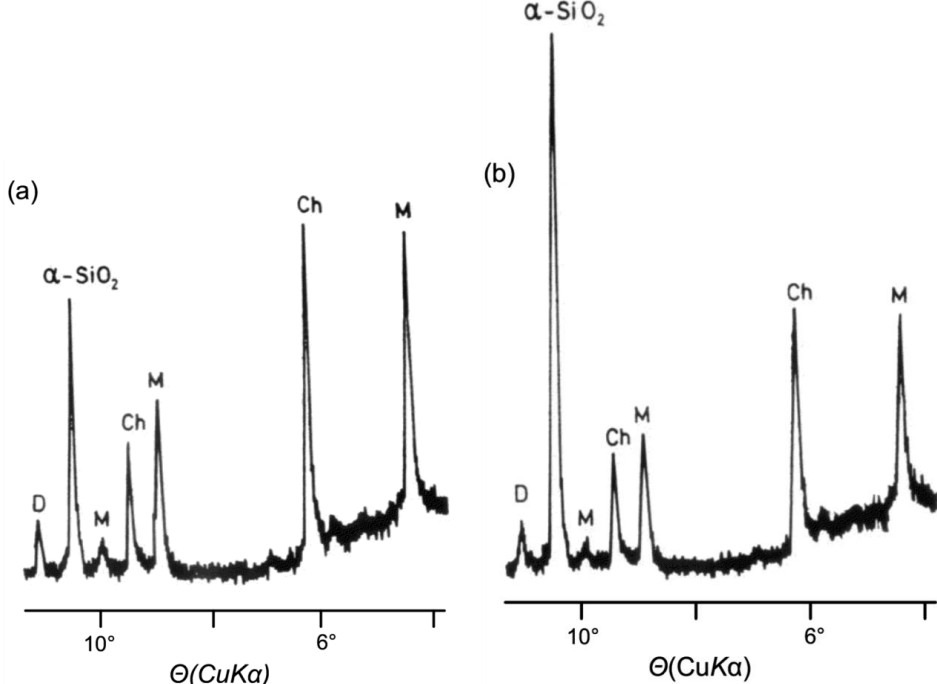

**Figure 1.** Parts of X-ray diffraction patterns of industrial dust; determination of the fraction of quartz, $\alpha$-SiO$_2$ ($X$) by the doping method (i): (**a**) original sample, (**b**) the sample doped with $X_a = 0.286$. The fraction of quartz in the original sample: $X = 0.200(10)$. Other phases: Ch, chlorites; D, dolomite; M, micas. Radiation CuK$\alpha$ [10].

### 4.2. Determination of the Fraction of the Dominant Phase

Let the system contain a phase, say $X$, which is dominant. Then, diffraction lines of other phases in the system are weak. The doping method (i) may not be appropriate in such a case, as diffraction lines of the phase used as the reference phase will be even weaker after doping. In this case three diffraction patterns are needed [7,8]:

- the diffraction pattern of the original sample, measuring $I_X$,
- the diffraction pattern of the sample doped by a known fraction, $X_a$, measuring $I_{X\,d+a}$,
- the diffraction pattern of the pure phase $X$, measuring $I_{X0}$.

It is important that all diffraction patterns are taken under the same conditions. The fraction of the phase $X$ in the original sample is given by equations:

$$X = (X_a/P) \, (R_1 \, / \, R_2), \, R_1 = 1 - (I_{X \, d+a}/I_{X0}), \, R_2 = (I_{X \, d+a}/I_X) - 1.$$

$P$ is the fraction of the original composition in the doped sample:

$$P = A_d + B_d + C_d + \ldots + X_d + Y_d + \ldots = 1 - X_a.$$

In order to increase the accuracy in determination of the fraction of the phase $X$, several diffraction lines of the phase $X$ can be used. However, this method is not applicable in a case where the fraction of $X$ is close to 1, as $R_1$ and $R_2$ then tend to zero and the value obtained for $X$ may not be reliable.

### 4.3. A New Application of the Doping Method

It has recently been proposed that the doping method (i) can be applied in study of decomposition of supersaturated solid solutions, and, in particular, of supersaturated intermetallic alloys [11–13]. For instance, the fraction of the precipitate $\beta$(Zn) formed during the aging of the supersaturated solid solution of Al–Zn alloys at high temperature (HT) can be estimated through these steps:

– quenching the alloy from HT to room temperature (RT), in order to stop or essentially slow down the decomposition process,
– doping the quenched alloy by a substance having the same (or very similar) chemical composition and crystal structure as the precipitate, that is, Zn instead of $\beta$(Zn) in case of Al–Zn alloys,
– taking into account a possible (although very slow) continuation of the decomposition process at RT.

Here is a description of the above steps, in short: Quenched sample:

$$A + B = 1, \, A = \alpha(M/\beta), \, B = \beta(Zn),$$

where $\alpha(M/\beta)$ is the matrix of the alloy, rich in Al, which is in a metastable equilibrium with precipitates $\beta$(Zn).

The sample doped by a known fraction, $B_a$:

$$A_d + B_d + B_a = 1.$$

The fraction of $B$ is given by (analogously to (8) and (9)):

$$B = B_a \, R_{AB}/(P - P \, R_{AB}), \, P = 1 - B_a, \, R_{AB} = (I_{A \, d}/I_{B \, d+a})(I_B/I_A).$$

The doping method can be also applied to colloid systems: (super)saturated liquid (water) solutions, in which the precipitation can be stopped or essentially slowed down.

### 4.4. Remarks on the Doping Methods

The appropriate analysis of the doping methods (i) and (ii) shows that it is advisable to choose $X_a$ close to 0.5 in order to achieve a better accuracy in the derived value of $X$ [6,7,10]. Grinding/mixing of the original and doped samples is necessary to ensure sample homogeneity, however, this may be a problem if the phases in the sample have rather different densities, grain sizes and shapes. In the case of the grains/crystallites being rather small, having the size of 1–10 μm, the primary extinction is small. The integrated intensities of diffraction lines should be precisely measured. The background line may be estimated by application of the appropriate procedures. It is advisable that the pure phase added to the original sample has a similar degree of crystal perfection as the same phase present in the original sample.

In application of the doping methods the preferred orientation of the grains/crystallites should be small. In case of its presence, the measured intensities of diffraction lines deviate from the true

values corresponding to the random orientation of grains/crystallites. The presence of the preferred orientation can be noticed if rather different values of the fraction of *X* are obtained for different combinations of diffraction lines of *X* and *Y*. Grains/crystallites having a plate-like or a needle-like shape tend to assume a preferred mode of orientation when mounted in the sample holder. If the degree of the preferred orientation is not high, a rather accurate value of *X* can be obtained by averaging the data that follow from diffraction lines which are differently affected, by application of the plot of $I_{X\,d+a}/I_{Y\,d}$ as a function of $(I_X/I_Y)$ (Equation (10)). If the preferred orientation of grains/crystallites is small, the fractions of the phases can be also found by comparison of the measured intensities of diffraction lines and the intensities of the same diffraction lines calculated on the basis of the crystal structure of the phases in question.

The doping method (i), described above, is based on the non-overlapping diffraction lines. If a phase, say *X*, exhibits several partially overlapped diffraction lines in a narrow angular interval, all these lines may be considered as a single diffraction line, if diffraction lines of other phases are not present in that interval. The problem of overlapping diffraction lines can be overcome by means of the individual profile fitting method which enables derivation of the profiles of particular diffraction lines. It has been suggested to combine the doping method (i) and the individual profile fitting method, thus performing the quantitative phase analysis without the reference to structural models of particular phases [9].

A detailed discussion on improving accuracy of QPA is given in [3] in sections: Minimizing Systematic Errors, Minimizing Sample-Related Errors, Crystallite-Size Issues, Preferred Orientation, Microabsorption, Whole-Pattern-Refinement Effects (pp. 364–370).

## 5. The Rietveld Approach

The contemporary experimental facilities and theory of XRD enable that the fraction(s) of the phase(s) present in the studied sample can be derived using the Rietveld method. That method is in essence a full pattern analysis technique. Models of the crystal structures of the phases present in the sample, together with instrumental and background information, are used to generate the theoretical diffraction pattern that can be compared to the observed pattern. The least-squares procedure is then used to minimize the difference between the calculated diffraction pattern and the observed diffraction pattern by adjusting the parameters of the model of the crystal structure. That procedure may result in determination of the fractions and microstructural parameters of the phases present in the sample and in refinement of their crystal structures [14]. The Rietveld-based QPA are concisely described in Reference [3] (pp. 348–350, 352–353).

## 6. Round Robin on Quantitative Phase Analysis

The IUCr Commission on Powder Diffraction organized a world-wide Round Robin on quantitative phase analysis, QPA, [15–17]. The particulars of the suggested procedures were:

– the types of analysis: measurement of integrated intensities, diffraction line profile fitting,
– the Rietveld method, application of different methods in QPA, the use of database of observed patterns, etc.
– sources: laboratory and synchrotron X-rays, neutron reactor radiation
– the aim: determination of phase fractions from diffraction data.

The expected results of the Round Robin were:

– to document powder diffraction techniques commonly applied in QPA
– to assess levels of accuracy, precision, and limits of detection
– to identify problems and suggest solutions
– to formulate recommended procedures for QPA using diffraction data
– to create a standard set of samples for future reference.

Round Robin included polyphase samples of various complexity:

- simple three-phase sample (corundum, fluorite, zincite)
- sample containing an amorphous phase
- sample with a phase showing preferred orientation
- sample exhibiting a problem of microabsorption
- complex synthetic and natural mineral phases
- pharmaceutical samples.

Main results of Round Robin are the following (in short) [15–17]:

- the major difficulty is caused by the lack of the operator expertise, which becomes more apparent with more complex samples
- some of these samples introduced the requirement for skill and judgement in sample preparation techniques
- a great obstacle to accurate QPA for X-ray diffraction based methods is the presence of absorption contrast between phases (microabsorption), which often cannot be solved.

## 7. Selected Studies with Application of Doping Methods

The doping methods have been applied in a series of author's studies, among others, in the study of the phase transition in titanium dioxide [18,19], biomineralization processes in the oyster *Ostrea edulis* [20,21], crystalline phases in intermetallic oxides [22–27], precipitation processes in Al–Zn alloys [11–13,19].

## 8. Phase Transitions in Titanium Dioxide

The phase transition of anatase, **A**, to rutile, **R**, does not take place at a fixed temperature, and the data found in the literature are rather contradictory. The temperature and kinetics of the transition **A→R** depend on the characteristics of the starting **A**, such as particle/crystallite size (specific surface), strain in the crystallites, the content and kind of impurities, deviation from stoichiometry, as well as on the atmosphere and pressure to which the material is exposed.

The phase transition **A→R** was followed by in situ X-ray diffraction. The experiments were performed with an as-synthesized $TiO_2$ of a high purity. Characteristic parts of XRD patterns of $TiO_2$ at RT, at 1573 K, and again at RT, after a complete heating and cooling cycle, are shown in Figure 2. The dominant phase at RT (before heating) was **A**, while **R** was present in traces. One can notice rather sharp diffraction lines due to big crystallites (estimated as 140–150 nm in size). During the heating run, the width of diffraction lines slowly increased due to increased thermal vibrations of atoms. At the same time, diffraction lines shifted toward smaller Bragg angles due to thermal expansion, which was found anisotropic. Above approximately 1200 K a gradual transition **A→R** took place. The nuclei of **R** were formed both at the surface and in the interior of the **A** crystallites, and these nuclei grew in number and size. At the highest reached temperature in this experiment, 1573 K, the fraction of **R** was bigger than that of **A**. During the cooling run, the transition **A→R** continued and apparently ceased below approximately 600 K. At RT, after cooling, **R** was the dominant phase, with several molar percent of **A**. **R** also exhibited the anisotropy in thermal expansion. The dependence of the molar fractions of **R** and **A** on temperature is shown in Figure 3. In this work it was concluded that the preferred orientation of crystallites in the sample was negligible. In the case of the phases **R** and **A,** the problem of absorption of X-rays can be neglected. Therefore, one can find the molar fractions of the phases **R** and **A** by comparison of the measured intensities and the intensities of diffraction lines calculated on the basis of the crystal structure of **R** and **A**, for instance, of the **R** diffraction line 110, and of the **A** diffraction line 101, according to $R/A = 1.25\ I_{110R}/I_{101A}$. At several temperatures, the molar ratio was also checked at RT by the doping method, after quenching the sample to RT [18,19].

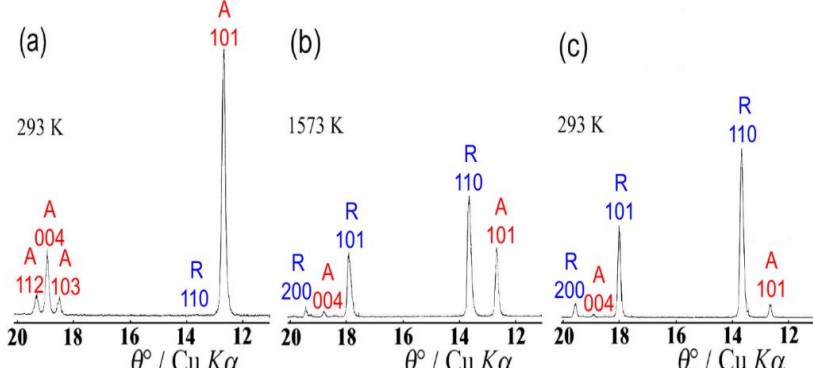

**Figure 2.** Characteristic parts of XRD patterns of the as-synthesized $TiO_2$ at (**a**) RT, (**b**) 1573 K, and (**c**) RT, after a complete heating and cooling cycle; **A** = anatase, **R** = rutile [19].

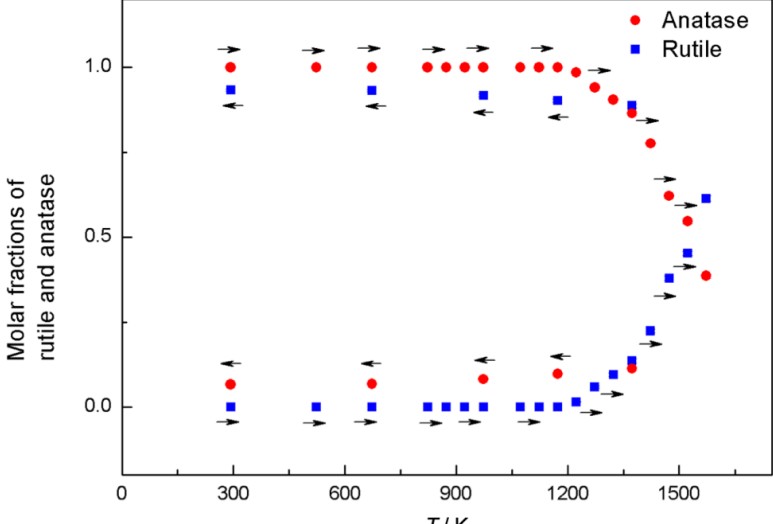

**Figure 3.** The dependence of molar fractions of anatase, **A**, and rutile, **R**, on temperature of the as-synthesized $TiO_2$ during the heating and cooling cycles. The arrows indicate the sense of the temperature change [19].

## 9. Biomineralization Processes in *Ostrea edulis*

Biomineralization is a combination of biochemical and physiological processes, which depend on the endogenous activity of the organism and the environmental influence. In bivalve molluscs, biomineralization starts at the early stage of the development and depends on various conditions: super-saturation of the medium with calcium ions, nature of the critical nucleus, the kind of organic matrix, inhibitors of phase transitions, environmental temperature, salinity, and pH. These processes take place in tissues and shells and are manifested in calcification and in polymorphic transitions of calcium carbonate in shell layers. The shells of marine bivalve molluscs are built of calcium carbonate in the form of calcite, aragonite or vaterite, which are distributed in two or more shell layers that differ in size, orientation, and type of crystal packing. The fractions of mineral components in carbonate shells depend on many factors, being a characteristic for family, genus, and bivalve species [20,21].

The biomineralization sequences with respect to the development stages, from the embryonal and larval to the juvenile oyster *Ostrea edulis* (sampled in the Limski Kanal, the Adriatic Sea, Croatia) was studied by XRD. All the development stages were followed, from the embryonic stage through the transition between the trochophore and veliger larva (prodissoconch I and prodissoconch II) and later, after swarming, the pelagic free-swimming larval stages, up to their settlement and attachment (from the D-shaped to the fully formed pediveliger larva), and finally during the metamorphosis and juvenile

stages (dissoconch). In the first gastrula stage, only an amorphous tissue is present (periostracum and organic matrix).The beginning of formation of the prodissoconch I is manifested by a small decrease of the organic tissue and by the presence of the first calcite crystals which act as crystallization centers for aragonite. In the later stage of the veliger larva the fraction of calcite decreases as well as the amorphous fraction, while the fraction of aragonite increases. A further increase of the fraction of aragonite (to approximately 0.95) suggests that prodissoconch II has been formed. The larvae are in the black phase, the incubation period is completed and the larvae are ready to swarm. The plankton period is followed by morphological changes of the prodissoconch II, which retains its unchanged mineral composition. Aragonite is dominant, the amorphous tissue is present in small fraction, while calcite is hardly detectable. The mature pediveliger larvae settle and attach to substrates. After the settlement the fraction of calcite increases. Metamorphosis starts when the clear border line between the prodissoconch II and the dissoconch is visible. The postmetamorphic shell of the juvenile oyster, as well of the adult oyster, consists mainly of calcite (with the fraction up to 0.95), except the resilium and myostracum which remain aragonitic, possibly as a continuation of the inner layer of the larval shell [20,21].

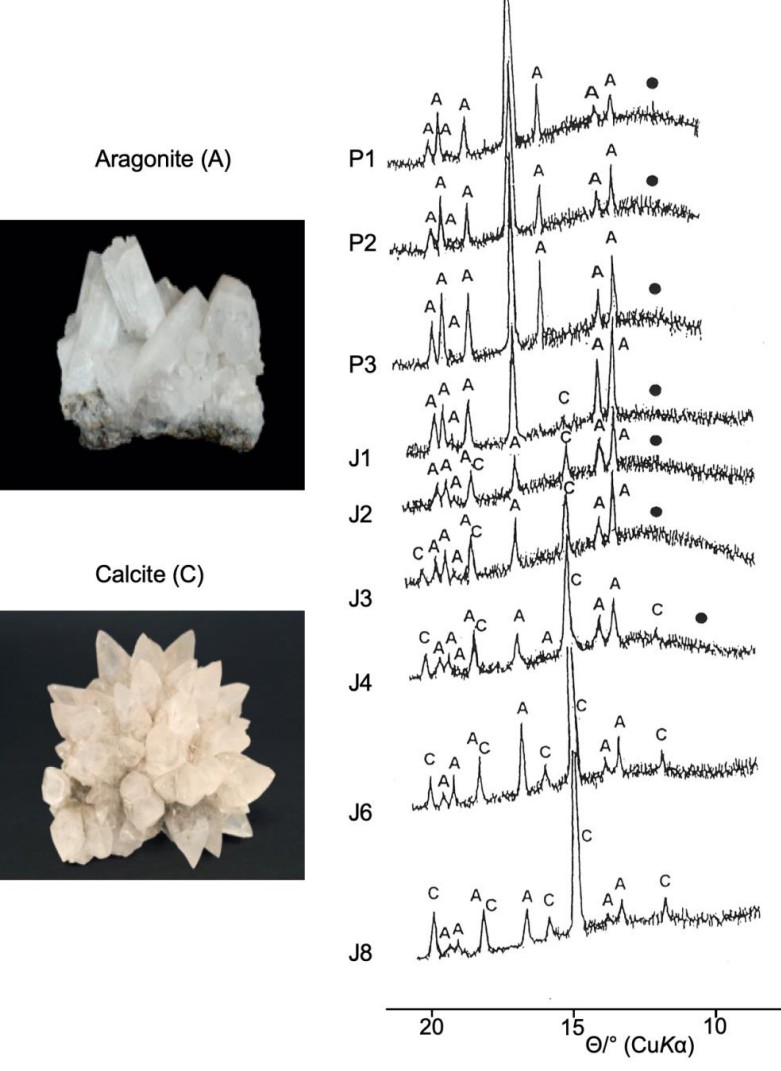

**Figure 4.** Characteristic parts of diffraction patterns of selected samples of *Ostrea edulis* of different development stages, P1–J8 (see Figure 5); C—calcite; A—aragonite [21].

The fractions of calcite and dolomite were deduced by the application of the doping method (i), by adding known fractions of calcite or aragonite to samples for which the development process (which

is otherwise slow) was stopped or essentially slowed down. Parts of diffraction patterns of selected samples of *Ostrea edulis* of different development stages and the dependence of mineral fractions on the development stages of *Ostrea edulis* are shown in Figures 4 and 5 [21].

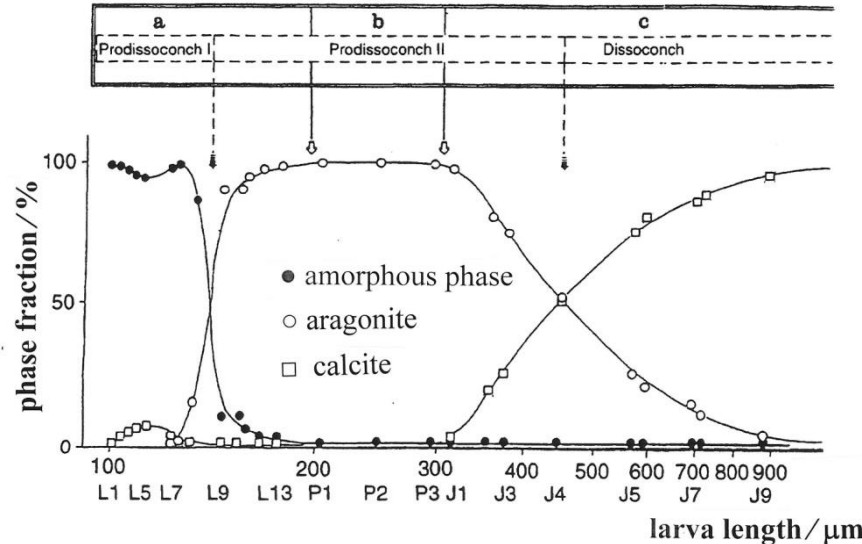

**Figure 5.** Variation of the fractions of amorphous tissue, aragonite, and calcite with the development stage in *Ostrea edulis*: multi-cell embryo (samples L1–L3), trochophore (L4, L5), early veliger (L6, L7), veliger/prodissoconch I (L8), veliger/prodissoconch I/II (L9), grey-black phase/prodissoconch II (L11, L12), black phase/prodissoconch II (L13), pediveliger/plankton (P1–P3), postsettlement/metamorphosis (J1, J2), prodissoconch II/dissoconch (J3), dissoconch (J4–J9) [21].

## 10. Crystalline Phases in Intermetallic Oxides

Intermetallic oxide systems e.g., ferrites, orthoferrites, alumina, zirconia, and titania—systems in which $Fe^{3+}$, $Al^{3+}$, etc. are replaced by other cations, have been systematically investigated by XRD, TEM, FE SEM, FTIR, and Mössbauer spectroscopies and other supplementary techniques (collaboration with S. Musić, M. Ristić, G. Štefanić, Ruđer Bošković Institute, Zagreb) in order to study formation of oxide phases and solid solutions, crystallinity, (nano)crystallite size, strain in the crystallites, properties at HT, phase transitions, and phase diagrams vs. physical and chemical properties. As an example of those studies, the systems $\alpha$-$Fe_2O_3$–$Ln_2O_3$, Ln = Eu, Gd, are described shortly [22–27].

Rare earth iron garnets, as well as substituted garnets, are important materials for advanced technology as they possess specific magnetic and magneto-optical properties. The studied samples were prepared by chemical coprecipitation and initial thermal treatment of corresponding hydroxides. Rare earth oxide, $Ln_2O_3$, reacts with iron (III) oxide, $\alpha$-$Fe_2O_3$, at high temperature to form orthoferrite with the perovskite structure: $Ln_2O_3 + \alpha$-$Fe_2O_3 \rightarrow 2LnFeO_3$. The rare earth orthoferrite, $LnFeO_3$, reacts with the additional $\alpha$-$Fe_2O_3$ to form the garnet-type ferrite: $3LnFeO_3 + \alpha$-$Fe_2O_3 \rightarrow Ln_3Fe_5O_{12}$. The phase composition, microstructure, and physical properties of the reaction products depend on the fractions of the initial reactants, the rare earth cations, temperature and other factors. The following phases appear in the systems $\alpha$-$Fe_2O_3$–$Ln_2O_3$: $\alpha$-$Fe_2O_3$ up to the fraction of the rare earth, $x(Ln) \approx 0.3$; $Ln_3Fe_5O_{12}$, being dominant at $x(Ln) \approx 0.4$; $LnFeO_3$, being dominant at $x(Ln) \approx 0.5$; $Ln_2O_3$ for $x(Ln)$ above $\approx 0.5$. No solid solutions were observed with certainty, even at the ends of the concentration range. Molar fractions of the observed crystalline phases (at RT), estimated by the application of the doping method (i), as a function of the initial molar fractions of $\alpha$-$Fe_2O_3$ and $Ln_2O_3$, are shown in Figure 6. Experimental procedures and obtained results, as well as the crystal data of the phases appearing in the systems $\alpha$-$Fe_2O_3$–$Ln_2O_3$ (at RT) are given in detail in References [22–27].

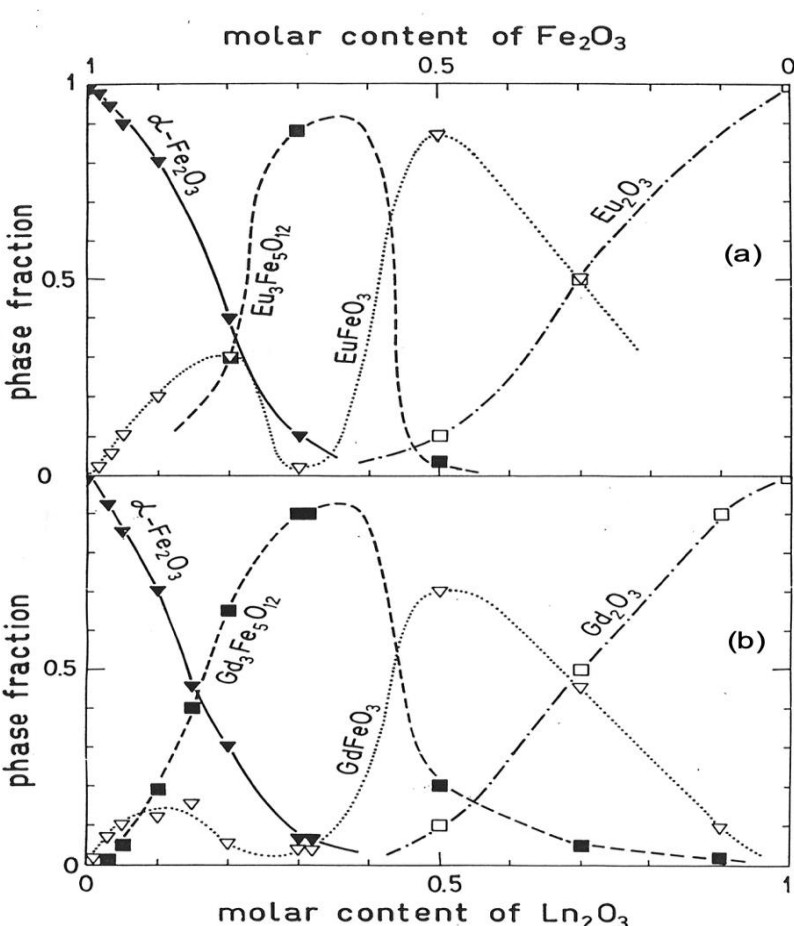

**Figure 6.** Approximate molar fractions (at RT) of the crystalline phases in the systems $\alpha$-$Fe_2O_3$–$Ln_2O_3$, Ln = Eu (**a**), Gd (**b**), as a function of the initial molar fractions of $\alpha$-$Fe_2O_3$ and $Ln_2O_3$. Samples were prepared at 1170 K [23].

## 11. Microstructure of Al–Zn Alloys

An extensive study of Al–Zn alloys, with the molar zinc content, $x(Zn)$, up to 0.62 by XRD, TEM, and DTA resulted in many new information on the microstructure of the alloys (unit-cell parameters, crystallite size and shape, strains, phase fractions, thermal expansion, etc.), in dependence on the initial composition, applied thermal treatment, and temperature [12,13,19]. Essential points from References [12,19] are given here. The alloys rapidly quenched from a temperature, $T_t$, higher than the solid-solution temperature, $T_{ss}$, to RT are supersaturated solid solutions. During aging, even at RT or at a higher temperature, the alloys decompose: the precipitation of cubic Guinier–Preston zones (GPZ) and hexagonal $\beta(Zn)$-phase (Figure 7), the transition of GPZ into $\beta(Zn)$ and the transition of the cubic $\alpha(M/GPZ)$-phase (in a metastable equilibrium with GPZ) to the cubic $\alpha(M/\beta)$-phase (in an equilibrium with the $\beta(Zn)$ precipitates) take place. The quenched alloys, being aged for a prolonged time at RT or at elevated temperature, as well as the alloys slowly cooled from $T_t$ to RT, approach the equilibrium state, in which $\alpha(M/\beta)$ coexists with the $\beta(Zn)$. The microstructures of the two groups of alloys are different. The slowly cooled alloys are much closer to the stable equilibrium state, than the quenched and aged alloys. In the latter alloys, residual strains in $\alpha(M/\beta)$ around the $\beta(Zn)$ precipitates are present. The quenched-in vacancies at RT are much more numerous in the quenched alloys than in the slowly cooled alloys.

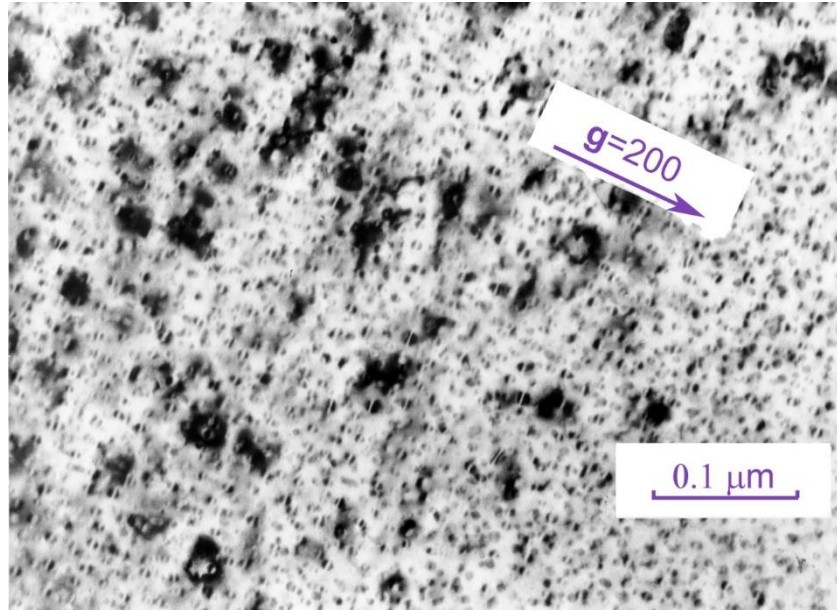

**Figure 7.** TEM photograph of an Al-rich alloy which was quenched from $T_t$ to RT and aged at 350 K for 1 day. A dense system of initial Guinier–Preston zones (GPZ) and initial β precipitates are visible. A line of zero diffraction contrast, normal to the diffraction vector, across the GPZ can be noticed [19].

Both quenched and aged alloys and slowly cooled alloys were gradually heated from RT to $T_t$, and their microstructure was followed in situ by XRD: enhanced thermal vibrations of atoms, thermal expansion, which is anisotropic for β(Zn), partial dissolution of β(Zn) into α(M/β), a change of the shape of the β(Zn) precipitates, phase transition of β(Zn) into the cubic α'-phase, or the transition of both β(Zn) and α(M/β) into α', coexistence of α' and α(M/α') phases, or of α', β(Zn) and α(M/β, α') phases, a change of composition of particular phases, and formation of solid solution, $α_{ss}$. $T_{ss}$ depends on the alloy composition as well as on the thermal treatment. On cooling the alloys from $T_t$ to RT, a temperature hysteresis is observed in reversal phase transitions. The microstructure at RT is different from the starting microstructure at RT, concerning the size and shape of the β(Zn) precipitates, their distribution in the crystallites of α(M/β), and the strains around the precipitates. During the repeated, second, heating and cooling runs, the alloys behave similarly as they do during the first cooling run. The changes of XRD pattern with temperature for the alloy with $x(Zn) = 0.48$ during the first and second heating and cooling cycles are illustrated in Figure 8. One can follow changes of the fractions of phases β(Zn), α(M/β), α', and α(M/β, α') with temperature.

This study confirms the fact that the quenched-in vacancies, vacancy-Zn pairs and other vacancy complexes play a dominant role in the diffusion rate of Zn atoms. The precipitation processes in the supersaturated solid solution depend on the balance of the quenched-in vacancies and traps for vacancies in the alloy subjected to a given thermal treatment [12,13,19].

It is of particular interest to follow the fraction change of the β(Zn) precipitates. In order to find out the fraction of β(Zn) at a given temperature, the alloy was rapidly quenched to RT (by a free fall into the water column in a vertical furnace, the estimated quenching rate being approximately $10^5$ K/s, [12]). As β(Zn) contains approximately 99.5 at. % Zn and approximately 0.5 at. % Al, the quenched alloy was doped with a known fraction of Zn and the doping method (i) was applied. It was found, for the alloy annealed at RT, that the fraction of β(Zn) varied in the interval from (approximately) the initial (fraction of Zn, $x(Zn)$, to (approximately) zero (at $T_{ss}$).

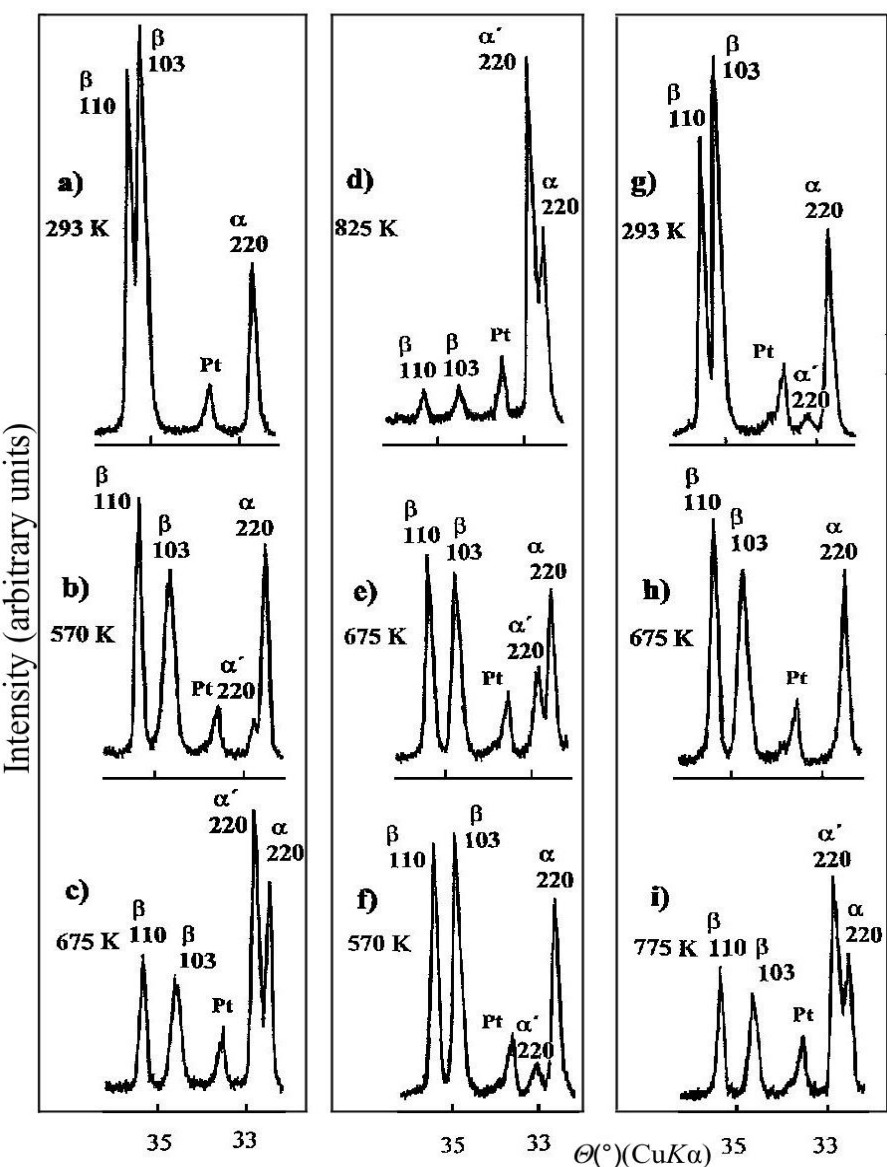

**Figure 8.** Characteristic parts of diffraction patterns at selected temperatures during the first and second heating and cooling cycles of the Al–Zn alloy with the initial content of Zn, $x$(Zn) = 0.48. The phases present in the alloy are: β(Zn) precipitates, α(M/β), α′, and α(M/β, α′). Prior to diffraction study at HT, the alloy was quenched from HT to RT and aged at RT for one week. Pt denotes the platinum heater of the sample [19].

## 12. Conclusions

In the present review, several well-known techniques of quantitative phase analysis by X-ray diffraction are concisely described. Special attention is paid to the doping methods. The intensity-fraction equations, used in the doping methods, deduced with no approximation, are free of the matrix effects—absorption of X-rays in the sample. It has been shown how to determine the fractions of several phases present in the sample from only two powder diffraction patterns, the one of the original sample and the other of the sample doped with known amounts of the phases of interest. It has been also shown how to apply the doping methods in determination of the fraction of the dominant phase, and, in a special case, how to determine the fraction of the amorphous content. The overlapping of diffraction lines can be solved by combining the doping method and the individual diffraction profile fitting method. The methods can be applied to a sample containing unidentified phase(s). Precautions

on how to obtain accurate and reliable results are discussed. Several examples of application of the doping methods are given, from the author's previous studies.

**Acknowledgments:** Experiments in this study were performed using the facilities located in the Ruđer Bošković Institute, 10000 Zagreb, Croatia, and in the Department of Physics, Faculty of Science, University of Zagreb, 10000 Zagreb, Croatia.

**Conflicts of Interest:** The author declares no conflict of interest.

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
