# Peer review of "Quantitative Phase Analysis by X-ray Diffraction—Doping Methods and Applications"

_crystals, doi:10.3390/cryst10010027_

Round 1

Reviewer 1 Report

In the manuscript several methods for quantitative phase analysis using XRD are briefly described. I consider that quite useful for a reader. I think, however, it would be even more useful if some methods and/or experimental procedures are described in a more detailed way: (1) "Chung's matrix flushing method" is described too shortly; (2) the footnote, that is "In the description of the software of PANALYTICAL ‒ X’PERT QUANTIFY it is stated: Addition (i.e. doping) models in which after the initial measurement on a sample, the sample is measured again with the concentration of the component of interest enriched by a known amount." is not quite clear to me.

(3) The sentence "Grinding/mixing of the original and doped samples is necessary to ensure sample homogeneity" is too general. I think in such a manuscript a more precise example of sample preparation could be given. Homogenous mixing two powders with different densities and possibly grain sizes is not easy.

(4) The manuscript ends too abruptly. Manuscripts usually have "summary" or "concusions" at the end - it should be added. 

Author Response

COVER LETTER   December 31, 2019

 crystals-683523

 Quantitative Phase Analysis by X-ray Diffraction – Doping methods, Applications

 Stanko Popović 1, 2

1Croatian Academy of Sciences and Arts, Zagreb, Croatia

2 Department of Physics, Faculty of Science, University of Zagreb, Zagreb, Croatia

I am sending a revised version of my review (now with a little modified title, according to a suggestion of one of the Reviewers): Quantitative Phase Analysis by X-ray Diffraction – Doping methods, Applications. (Manuscript ID: crystals-683523; Type of manuscript: Review; submitted on Dec 16, 2019), for the Special Issue "Crystal Structure Characterization by Powder Diffraction", Guest Editors Dr. Angela Altomare and Dr. Rosanna Rizzi, Institute of Crystallography, National Research Council-CNR, Bari, Italy.

   I would like to express my gratitude to both Reviewers for their very useful and valuable comments. I have accepted all the comments, trying my best to include them in the revised version of my review. A short Conclusion has also been added. The changes in relation to the as-submitted version (Dec 16, 2019) have been done using the Track Changes function in Microsoft Word: insertions shown in blue, deletions strikethrough-highlighted in bright green.

   I hope that I have consistently followed the Instructions to Authors in the preparation of my review. I have prepared my review on the basis of my previous papers and several papers of other authors. I have obtained permissions to quote parts of texts and figures from journals in which I have published my previous papers: Croatica Chemica Acta, Marine Biology, Acta Chimica Slovenica (the successor of Vestnik Slovenskega kemijskega društva) and Macedonian Journal of Chemistry and Chemical Engineering. All original publications are consistently cited throughout the review. Concerning this matter, I am grateful to Ms Eirland Zhang, Assistant Editor, who sent me (Dec 30, 2019) the following explanation: “According to our policy, 1. short quotes from a previously published article should be set off in quotation marks and original version cited; 2. permission must be requested when large sections are reproduced. For your manuscript, since you have already obtained the copyright permission and cite previous work, it is acceptable now. For common expressions, accepted by international scientific community, there is no need to rephrase.“

   I confirm that neither this review nor any part of its content is currently under consideration in another journal.

Kind regards and all the best in 2020

Stanko Popović

Professor Emeritus Stanko Popović

Full Professor in Physics (retired), Professor Emeritus

Department of Physics, Faculty of Science, University of Zagreb,

Bijenička cesta 32, POB. 331, 10002 Zagreb, Republic of Croatia

Fellow of the Croatian Academy of Sciences and Arts,          

Croatian Crystallographic Association (Secretary 1992-2006, Chair 2006-2016)

Trg Nikole Šubića Zrinskog 11, 10000 Zagreb, Republic of Croatia                                     

phone   ++385 1 460 5549; ++385 1 383 5006; ++385 1 489 5172

fax       ++385 1 468 0336; ++385 1 481 9979

e-mail   [email protected] 

web http://info.hazu.hr/hr/clanovi_akademije/osobne_stranice/stanko_popovic

Reviewer 2 Report

This is a review article of the author mainly about the author's works. In fact, it is highlighted that 18 out of 26 (69% of the cited references) are from the author. The abstract reflects the content of the review article but the title does not reflect the content of the article. As the scope of the work is within the journal, I advice acceptance of the work subjected to the address of the point raised next.

Major:
1. Please modify the title to better reflects the content of the paper. I suggest to remove "new perspectives" and to add "doping method" or similar.

2. The work(s) in the new book: Int. Tables for Crystallography volume H should be referenced.

3. The Rietveld approach section contains only one reference. This is the most used method for QPA. A bit more information here would be advisable.

Minor points:

Line 38 "weight fraction" instead of fraction. Line 49, it should be mentioned that specimen dependent effect() could also affect: like preferred orientation or extinction Line 92 ".... and in determination of the fraction of the amorphous content." Please elaborate this statement. Line 245. "... TiO2 p.a." Please describe better. It is not easy to understand. Line 346. "...new information on the microstructure of the alloys,.." Please elaborate in the microstructure applications.  

Author Response

(The authors gave the same response as above.)

Round 2

Reviewer 1 Report

no comments